# Low-Density Lipoprotein Receptor Is a Key Driver of Aggressiveness in Thyroid Tumor Cells

**DOI:** 10.3390/ijms241311153

**Published:** 2023-07-06

**Authors:** Giovanna Revilla, Lara Ruiz-Auladell, Núria Fucui Vallverdú, Paula Santamaría, Antonio Moral, José Ignacio Pérez, Changda Li, Victoria Fuste, Enrique Lerma, Rosa Corcoy, Fabián Pitoia, Joan Carles Escolà-Gil, Eugènia Mato

**Affiliations:** 1Institut de Recerca de l’Hospital de la Santa Creu i Sant Pau, Institut d’Investigació Biomèdica (IIB) Sant Pau, 08041 Barcelona, Spain; grevilla@santpau.cat (G.R.);; 2Department of Biochemistry and Molecular Biology, Universitat Autònoma de Barcelona (UAB), 08025 Barcelona, Spain; 3Department of Endocrinology and Nutrition, Hospital de la Santa Creu i Sant Pau, 08041 Barcelona, Spain; 4Department of General Surgery, Hospital de la Santa Creu i Sant Pau, 08041 Barcelona, Spain; 5Department of Medicine, Universitat Autònoma de Barcelona (UAB), 08193 Bellaterra, Spain; 6Department of Pathological Anatomy, Hospital de la Santa Creu i Sant Pau, 08041 Barcelona, Spain; 7CIBER de Bioingeniería, Biomateriales y Nanomedicina (CIBER-BBN), 28029 Madrid, Spain; 8Division of Endocrinology, Hospital de Clínicas, University of Buenos Aires, Buenos Aires C1120 AAF, Argentina; 9CIBER de Diabetes y Enfermedades Metabólicas Asociadas (CIBERDEM), 28029 Madrid, Spain

**Keywords:** thyroid cancer (TC), low-density lipoprotein receptor (LDLR), BRAF V600E, RAS/RAF/MAPK (MEK)/ERK pathway, low-density lipoprotein (LDL)

## Abstract

We previously described the role of low-density lipoprotein (LDL) in aggressiveness in papillary thyroid cancer (PTC). Moreover, the MAPK signaling pathway in the presence of BRAF V600E mutation is associated with more aggressive PTC. Although the link between MAPK cascade and LDL receptor (LDLR) expression has been previously described, it is unknown whether LDL can potentiate the adverse effects of PTC through it. We aimed to investigate whether the presence of LDL might accelerate the oncogenic processes through MAPK pathway in presence or absence of BRAF V600E in two thyroid cell lines: TPC1 and BCPAP (wild-type and BRAF V600E, respectively). LDLR, PI3K-AKT and RAS/RAF/MAPK (MEK)/ERK were analyzed via Western blot; cell proliferation was measured via MTT assay, cell migration was studied through wound-healing assay and LDL uptake was analyzed by fluorometric and confocal analysis. TPC1 demonstrated a time-specific downregulation of the LDLR, while BCPAP resulted in a receptor deregulation after LDL exposition. LDL uptake was increased in BCPAP over-time, as well as cell proliferation (20% higher) in comparison to TPC1. Both cell lines differed in migration pattern with a wound closure of 83.5 ± 9.7% after LDL coculture in TPC1, while a loss in the adhesion capacity was detected in BCPAP. The siRNA knockdown of LDLR in LDL-treated BCPAP cells resulted in a p-ERK expression downregulation and cell proliferation modulation, demonstrating a link between LDLR and MAPK pathway. The modulation of BRAF-V600E using vemurafenib-impaired LDLR expression decreased cellular proliferation. Our results suggest that LDLR regulation is cell line-specific, regulating the RAS/RAF/MAPK (MEK)/ERK pathway in the LDL-signaling cascade and where BRAF V600E can play a critical role. In conclusion, targeting LDLR and this downstream signaling cascade, could be a new therapeutic strategy for PTC with more aggressive behavior, especially in those harboring BRAF V600E.

## 1. Introduction

Thyroid cancer (TC) is the most common malignancy of the human endocrine system, with papillary thyroid carcinoma (PTC) being the most frequent histological subtype. The incidence of PTC has sharply increased in the last decade, accounting for 90% of all thyroid malignancies [1]. Despite most PTC having an excellent response to treatment after initial therapy, some subtypes may present with features that may determine a high risk of recurrence and even advanced disease, which eventually may become radioiodine refractory (RAI) [2].

The MAPK pathway is known for being critical in the development of many oncogenic diseases, such as melanoma, colon and breast cancer [3,4,5]. Various histological subtypes of TC present a constitutive activation of MAPK signaling pathway. RAS, RAF, MEK and ERK are the most relevant proteins in this pathway, and they are involved in cellular processes such as differentiation, proliferation and apoptosis [6]. Different studies of thyroid tumors have identified the activating mutation of BRAF as the main genetic alteration of this pathway, with the BRAF V600E hotspot mutation being the most frequently found genetic event. This oncogene is proposed to be the main early mutational event, occurring in about 50% of PTC diagnostics, and is consistently related to adverse prognostic evolution in tumors (>2 cm) [7,8]. BRAF V600E-driven tumors present high extracellular signal-regulated kinase phosphorylation, promoting unregulated cell proliferation and the inhibition of RAI responsive genes in TC through an aberrant activation of the MAPK pathway [9,10]. A recent report demonstrated a positive association between the presence of BRAF mutation, obesity and a greater risk of TC in tumors from patients with PTC [10]. 

Epidemiologic and experimental evidence have associated obesity/overweight and cholesterol levels as a risk factors of worse cancer progression, including TC [11,12,13,14,15]. Pre-clinical studies reported a higher low-density lipoprotein receptor (LDLR) expression in some malignant tumors such as colon cancer where low-density lipoprotein (LDL) increased the proliferation of these tumor cells through the MAPK signaling pathway [16]. 

Moreover, our group reported that cholesterol metabolite 27-hydroxycholesterol (27-HC) was accumulated in PTCs, promoting their aggressive behavior [17]. In this sense, we remark the association described between the LDLR expression and the abnormal lipidic metabolism as a factor of cancer progression and recurrence in hepatocellular carcinoma, lung cancer, breast cancer, colorectal cancer and prostate cancer [5,18]. The LDLR is a cell surface glycoprotein that plays an important role in regulating cholesterol homeostasis [19]. Elevated LDLR expression and LDL uptake in a wide range of tumors have been related to LDL-mediated cancer growth in mice [20,21]. Studies on pancreatic cancer elucidated that the blocking of LDLR reduces the proliferative and clonogenic potential of tumoral cells and decreases the activation of the ERK1/2 survival pathway sensitizing cells to chemotherapeutic drugs [22]. 

However, the potential molecular mechanisms that regulate this association in TC remain poorly understood. Different studies have established that the degree of MAPK signaling pathway activation determines the extent of LDLR transcription [5,23,24]. In this study, we aim to investigate whether BRAF V600E and LDL can potentiate the LDLR-mediated oncogenic processes via the RAS/RAF/MAPK (MEK)/ERK pathway.

## 2. Results

### 2.1. Differential Regulation of Low-Density Lipoprotein Receptor (LDLR) Protein Expression and Low-Density Lipoprotein (LDL) Uptake in Papillary Thyroid Carcinoma (PTC) Cell Lines with Different Mutational Status after LDL Incubation

To examine the protein expression kinetics of LDLR in response to LDL, TPC1 and BCPAP cells (wild-type and BRAF V600E, respectively) were treated under basal conditions (5% FBS), as a control, or with LDL (200 μg/mL ApoB) and harvested at 24 h, 48 h and 72 h to analyze LDLR protein expression via Western blotting. Figure 1A shows that LDL promoted a significant decrease in LDLR expression in TPC1, in comparison to BCPAP, where LDLR expression was maintained over the time. To determine the beginning of LDLR declining in TPC1 incubated with LDL, a time course at different times up to 12 h was carried out. In the initial 12 h period of exposure to LDL, TPC1 exhibited higher expression of LDLR protein, as it is demonstrated in Appendix A. However, subsequently, as depicted in Figure 1A, there was a gradual decline in expression levels until reaching a significant threshold, albeit not being completely inhibited. To study the LDL uptake, LDL was labeled with 19-dioctadecyl-3,3,39,39-tetramethyl indocarbocyanine (DiI) for subsequent analysis using fluorescence spectrophotometry and confocal microscopy. As illustrated in Figure 1B and Figure 2A, both cell lines were capable of acquiring DiI-LDL but displayed distinct uptake patterns. BCPAP demonstrated a progressive and higher uptake levels of DiI-LDL over time, in comparison to the TPC1 cell line, whose DiI-LDL uptake levels were significantly lower. Furthermore, considering the graph results of DiI-LDL uptake related to LDLR expression in TPC1, it is important to take into consideration the remnants of DiI-LDL acquired during the initial 12 h period, in which LDLR expression was higher, as well as the not complete abolition of LDLR expression at 24 h, 48 h and 72 h. Taking into account that aggregated LDL and oxidized LDL can be acquired through other receptors, such as LRP1 or LOX1, respectively, we initially tested whether the LDL showed any evidence of oxidation or aggregation (Appendix A).

Confocal analysis demonstrated lysosome and LDL colocalization in both cell lines (Figure 2A,B). However, the cell lines displayed different intracellular distributions, with lysosomes being broadly distributed in the cytoplasm in the TPC1 cells, while in the BCPAP cells, these organelles were mainly localized throughout the perinuclear region (Figure 2A). 

### 2.2. LDL Promotes Higher Proliferation Levels and Decreases Adhesion Capacity in the BCPAP Cell Line, in Comparison to the Constant Proliferation and Cellular Migration Promoted by LDL in the TPC1 Cell Line

To examine whether LDL regulates cellular proliferation and migration, TPC1 and BCPAP cell lines were maintained under basal conditions (5% FBS) as a control and were compared to cells incubated with LDL (200 μg/mL ApoB) for 24 h, 48 h and 72 h. As shown in Figure 3, LDL promoted cellular proliferation in both cell lines. However, in BCPAP cells, the percentage of proliferation was significantly higher over the time compared to TPC1 cells, which also displayed an increase in proliferation but to a lesser extent (127.1% ± 9.30; 181.0% ± 13.01; and 169.54% ± 28.43 in BCPAP cells incubated with LDL for 24 h, 48 h and 72 h, respectively, vs. 113.7% ± 7.84; 127.9% ± 11.86; and 156.4% ± 14.00 in TPC1 cells). These findings are consistent with the amount of internalized DiI-LDL observed in both cell lines (Figure 1B).

To study the effect of LDL on cell migration, the wound-healing assay was performed by incubating TPC1 and BCPAP cell lines under basal conditions (5% FBS) or with LDL (200 μg/mL ApoB) for 16 h. The results presented in Figure 4 revealed significant differences between both cell lines. In the TPC1 cell line, the wound-repair percentage was 83.47% ± 2.36 compared to basal conditions (5% FBS), which was 22.35% ± 3.08 (Figure 4A). However, in the BCPAP cell line treated with LDL, there was a loss of cellular adhesion with a higher percentage of cell suspension (39.63% ± 2.14) compared to the control (16.80% ± 1.41, Figure 4B). Regarding the cell viability of these unanchored BCPAP cells under LDL treatment, trypan blue staining demonstrated 56.33% ± 4.33 viability compared to unanchored BCPAP cells maintained under basal conditions (29.50% ± 6.71, Appendix A). 

### 2.3. LDL Enhances the RAS/RAF/MAPK (MEK)/ERK Pathway 

As a next step towards understanding the mechanism of action of LDL in both cell lines, TPC1 and BCPAP, we analyzed the most relevant proteins of both PI3K-AKT and RAS-RAF-MAPK (MEK)/ERK pathways, two critical signaling pathways related to proliferation and migration, after LDL incubation (200 µg/mL ApoB) for 24 h, 48 h and 72 h.

In TPC1 cells, a moderate increase in p-ERK expression was observed after 24 h of LDL exposition in comparison to basal conditions (5% FBS), but without reaching significant levels, and followed by a progressive decrease over the time. Otherwise, no changes were detected in ERK expression when compared to basal conditions (5% FBS). Additionally, AKT expression was increased after LDL treatment but without reaching significant levels. Although p-AKT and p-MTOR levels did not suffer statistical modifications after LDL treatment in comparison to basal conditions (5% FBS), p-MTOR displayed a slight tendency to decrease at 48 h and 72 h after LDL exposure (Figure 5A). On the other hand, in BCPAP cell line, there was a significant increase in p-ERK expression after LDL incubation compared to basal conditions (5% FBS); otherwise, there were no changes in ERK expression. Moreover, AKT expression increased over time after LDL exposure, in comparison to basal conditions (5% FBS). In contrast, cells incubated under basal conditions (5% FBS) demonstrated a significant increase in p-AKT, but only at 48 h. Finally, no changes were observed in terms of MTOR and p-MTOR over the time (Figure 5B). Interestingly, p-ERK expression was increased in both cell lines, particularly in the BCPAP cell line, demonstrating that LDL promoted a significant overactivation of the RAS/RAF/MAPK (MEK)/ERK pathway in the cell line harboring BRAF V600E (BCPAP). 

### 2.4. Vemurafenib Reduces Cellular Proliferation Promoted by LDL, Downregulating LDLR Expression and Impairing RAS/RAF/MAPK (MEK)/ERK Pathway Activation in BCPAP Cell Line

As shown in Figure 6, BCPAP cells were treated with vemurafenib (1 μM) with/without LDL (200 μg/mL ApoB) and were compared to the LDL-only treated condition at 24 h, 48 h and 72 h. Protein expression analysis revealed an increase in LDLR and p-ERK expression after LDL exposure, according with previous findings depicted in Figure 1A and Figure 5B. Additionally, the treatment with vemurafenib by itself produced a significant decrease in both p-ERK and LDLR protein expression. Importantly, vemurafenib completely counteracted the LDL-mediated effects on LDLR and p-ERK expression (Figure 6A,B). 

To assess the effects of vemurafenib on LDL-induced proliferation, the BCPAP cell line was treated with the same conditions described above and the proliferation was analyzed via MTT assay. As expected, incubation with LDL resulted in an increase in cellular proliferation compared to basal conditions (Figure 6C), as it has been previously reported in Figure 2. Vemurafenib also blocked the LDL-mediated increases in cell proliferation, reaching similar proliferation levels as the basal conditions (5% FBS, Figure 6C). Conversely, in the TPC1 cell line treated with LDL (control cell line), vemurafenib did not produce further changes in either LDLR expression or ERK phosphorylation or proliferation (Appendix A).

### 2.5. LDLR Is Required for LDL-Mediated Induction of RAS/RAF/MAPK (MEK)/ERK Pathway, LDL-Uptake and Cell Proliferation in the BCPAP Cell Line

In order to investigate whether LDL has a role as a coadjuvant with BRAF V600E in the cell proliferation, BCPAP cells were silenced for 48 h with a LDLR siRNA (siLDLR) before LDL treatment, resulting in an around 60% to 70% decrease in LDL uptake, LDLR gene expression, as well as LDLR protein expression (Figure 7A–C).

Furthermore, as shown in Figure 7D, the knockdown of LDLR in BCPAP cells exposed to LDL resulted in a significant reduction in p-ERK expression compared to the LDL-alone treatment. In addition, LDL did not further upregulate p-ERK expression after silencing LDLR, suggesting that LDLR is the main driver of the LDL-mediated induction of the RAS/RAF/MAPK (MEK)/ERK pathway (Figure 7D). Interestingly, the LDLR knockdown in combination with vemurafenib treatment in BCPAP cells exposed to LDL resulted in a somewhat greater p-ERK protein expression decrease, implying a potential synergistic effect between LDLR and BRAF V600E. Otherwise, there were no changes in ERK expression (Appendix A). In line with these findings, LDLR knockdown also impaired cell proliferation in BCPAP cells exposed to LDL (Figure 7B,E). Further impaired cell proliferation was found when LDLR was knockdown in cells treated with vemurafenib (Figure 7E). 

## 3. Discussion

Some epidemiological studies have demonstrated a relationship between obesity/overweight, cholesterol and a higher incidence rate, as well as worse prognosis, of certain solid tumors [11,12,13,14,15]. Moreover, it has been described the potential therapeutic effect of targeting lipid metabolism using classic lipid-lowering drugs, such as statins, in cancer therapy to prevent LDL effects on tumor progression [25]. Although the link between lipid metabolism and tumor aggressiveness is complex and not yet fully understood, several reports have identified these connections in certain solid tumors. Furthermore, there is evidence that the oncogenes and tumor suppressor genes have important roles as the drivers of alterations in lipid metabolism in cancer [17,26,27,28,29,30].

In TC, we previously identified a decrease in LDL levels in the serum of patients with a more aggressive histological pattern, whereas in their tumoral tissue, there was an upregulation in the expression of the *LDLR* and a decrease in the *3-hydroxy-3-methylglutaryl-CoA reductase (HMG-CoA)* gene expression, responsible for the biosynthesis of cholesterol. Moreover, we also observed an intratumoral increase in the 27-HC metabolite together with a downregulation of the 25-HC 7-alpha-hydroxylase (CYP7B1) enzyme that controls its degradation [17]. In this line, it is known that the most common histological pattern of well-differentiated TC corresponds to PTC, and the subtype with aggressive behavior and worse prognosis in patients is associated with the presence of the BRAF V600E mutation [31,32,33]. The BRAF V600E belongs to the RAS/RAF/MAPK (MEK)/ERK pathway, which is the most important signaling cascade related with proliferation, differentiation and apoptosis, and promotes ERK hyperactivation that is critical in cancer development and progression [34]. Moreover, LDL was shown to activate the RAS/RAF/MAPK (MEK)/ERK pathway, promoting oncogenic processes in tumors such as colorectal cancer, as well as in endothelial dysfunction in atherosclerosis [35,36].

Considering the current evidence of the association between LDL and tumor progression, and the effect of LDL on MAPK activation [37], we aimed to study the role of LDL in PTC cells and the connection with BRAF V600E mutation as a metabolic trigger of the tumor aggressiveness. Thus, we analyzed the LDL-associated uptake through the LDLR, as well as the signaling pathways related to the presence or absence of the BRAF V600E mutation in two TC cell lines: TPC1 and BCPAP.

The most important findings of our study show that the LDLR expression and LDL uptake were differentially regulated in the presence versus the absence of the BRAF V600E mutation, with the BCPAP cell line showing poor LDLR regulation. Moreover, due to this LDLR expression in BCPAP cell line, the LDL uptake was significantly higher in comparison to TPC1. On the other hand, the amount of LDL uptake in TPC1 cells was maintained over the time due to a not complete inhibition of the LDLR expression in the cell membrane, as well as the LDL particles taken up during the first 12 h of LDL exposure, where LDLR expression was higher (Appendix A). In addition, the BCPAP cell line displayed a higher proliferation percentage and an increase in the loss of cellular adhesion compared to the TPC1 cell line. Similar independent studies were performed on anaplastic thyroid cancer (ATC) cell lines, CAL-62 and 8505C (wild type and harboring BRAF V600E, respectively), exposed to LDL conditions and showed comparable results regarding LDLR protein expression, LDL uptake, cellular proliferation and MAPK signaling pathway (Appendix A).These data are in accordance with previous studies that demonstrates LDL capacity to induce the proliferation, migration and loss of adhesion in breast cancer cells [38]. Nevertheless, these differences between both cell lines after LDL treatment could be related to its different mutational status. Different reports highlight how lipid metabolic dysregulation due to high systemic cholesterol uptake could lead to an important metabolic alteration in tumoral processes. This supports the suggestion that cholesterol plays a critical role in tumor metastatic promotion with the presence of the BRAF mutation [39,40,41,42,43,44]. 

Moreover, the lysosomal distribution pattern differed between both cell lines. The lysosome distribution of BCPAP cell line was more juxtanuclear; meanwhile, in the TPC1 cell line, the lysosomes were distributed throughout the cytoplasm. These results may suggest a higher interaction with the endoplasmic reticulum (ER) and Golgi in BCPAP cell line, in comparison to TPC1, as a consequence of different metabolic signaling inputs due to their differential capacity for LDL uptake [45]. It is well known that the LDL uptake is released to the lysosomes for their degradation, as well as the delivery of free cholesterol. However, our findings are in line with the newly reported roles of lysosomes in relation to tumoral cells, indicating a potential connection with other important organelles, such as the ER, mitochondria and nucleus, which are crucial in the cholesterol metabolism of cancer cells [45,46]. The lysosomal system has emerged as an important factor in invasion, dissemination and survival related to oncogenic processes [47]; for that reason, further investigation should be considered to clarify its role in TC progression. 

To elucidate the signal transduction pathway, through which LDL promoted the proliferation, adhesion and migration processes in TC cell lines, we studied the most relevant transduction pathways involved in these processes in TC, namely the PI3K-AKT and RAS-RAF-MAPK (MEK)/ERK signaling cascades. In accordance with Velarde V. et al., who established a crosstalk between native LDL and ERK phosphorylation through the MAPK pathway in vascular smooth muscle cells [48], our results also demonstrated an increase in p-ERK expression after LDL exposure. However, the BCPAP cell line demonstrated a greater increase in ERK phosphorylation, suggesting the hyperactivation of the RAS/RAF/MAPK (MEK)/ERK pathway due to its higher levels of LDL uptake. These results were also observed in the ATC cell line, 8505C, harboring the BRAF V600E mutation (Appendix A). Therefore, the RAS/RAF/MAPK (MEK)/ERK pathway could be considered an LDL-related signaling pathway. On the other hand, regarding the PI3K-AKT pathway, both cell lines, TPC1 and BCPAP, resulted in a higher phosphorylation of AKT when cells were nutrient-deprived (5% FBS) in comparison to LDL conditions, being significant in BCPAP cell line. These findings are directly in line with previous studies in cancer cells that suggest an activation of PI3K-AKT by increasing AKT phosphorylation to survive stressful environments promoted by serum starvation [49,50].

Importantly, we also investigated whether BRAF V600E partial suppression through vemurafenib decreases its oncogenic effects by modulating LDLR expression and MAPK signaling cascade in BCPAP cell line. We found that BRAF V600E-mediated oncogenic signaling was partially blocked by vemurafenib in the BCPAP cell line treated with LDL, causing a decrease in the p-ERK signaling pathway. Consequently, proliferation decreased, as has also been reported by Xing et al., in TC cell lines [51]. Interestingly, we observed a link between the RAS/RAF/MAPK (MEK)/ERK and LDL signaling pathways when vemurafenib was able to induce a strong LDLR downregulation, as well as to attenuate cell proliferation promoted by LDL exposure in the BCPAP cell line. These results show, for the first time to our knowledge, that vemurafenib could also regulate the recycling or gene transcription of *LDLR* in the BCPAP cell line. Otherwise, no significant changes were found, either in LDLR or p-ERK in the TPC1 cell line used as a control, and the cytotoxic effects of this drug can be discarded. Moreover, the combination of siRNA-mediated targeting *LDLR* plus vemurafenib in LDL-treated conditions can trigger a significant reduction in the LDL-induced ERK phosphorylation, as well as cellular proliferation, suggesting a synergistic effect between LDL and BRAF mutation in TC. In this sense, similar results in regard to a synergic effect between LDL-signaling and BRAF V600E were described in melanoma, colorectal and lung cancer cells [5,52,53,54]. Nevertheless, further research is needed to better understand the role of vemurafenib in lipid metabolism.

In summary, our findings support the suggestion that LDLR plays an important role in the RAS/RAF/MAPK (MEK)/ERK signaling cascade in TC and suggest a synergy between LDL-mediated receptor uptake and BRAF V600E in TC, which could lead to a worse prognosis in the clinical setting of hypercholesterolemia (Figure 8). Further investigation is needed in terms of using cholesterol-lowering drugs in combination with RAS/RAF/MAPK (MEK)/ERK inhibitors as a possible therapeutic strategy for PTC with aggressive behavior.

## 4. Materials and Methods

### 4.1. Cell Lines and Cell Culture

The experiments were carried out on cell lines derived from human PTC, TPC1 (bearing RET/PTC rearrangement) and BCPAP (bearing the BRAF V600E oncogene). Both cell lines were provided by Paolo Vigneri of Azienda Ospedaliero Universitaria Policlinico Vittorio Emanuele Catania, Catania, Sicilia, IT. Cells were cultured in RPMI 1940 medium (ThermoFisher Scientific, Waltham, MA, USA) supplemented with 10% FBS, 100 U/mL penicillin and 1 μg/mL streptomycin at 37 °C in a 5% CO_2_ atmosphere.

### 4.2. Human LDL Isolation

Human LDL (1.019–1.063 kg/L) was isolated via the sequential ultracentrifugation of fasting plasma. ApoB levels were determined enzymatically and by immunoturbidimetric assays, respectively, applying commercial kits adjusted to a COBAS c501 autoanalyzer (Roche Diagnostics, Minato City, Tokyo) [55,56]. Human LDL particles were not oxidatively modified during their isolation and the experimental procedure. The oxidative modification of human LDL was measured via the monitoring of the formation of conjugated dienes at 234 nm at 37 °C with a BioTek Synergy HT spectrophotometer (BioTek Synergy, Winooski, VT, USA). As a positive control, LDL was also oxidized by adding CuSO_4_ (2.5 mol/L) to wells containing LDL (0.1 mg of apoB/mL). At the end of the process, an aliquot of native LDL oxLDL and the substance oxidized for 2 h (partially oxidized LDL) were stained with Sudan Black and run on an agarose gel (0.5%) for 40 min to evaluate their integrity and charge properties (Appendix A). 

### 4.3. Labeling LDL with DiI

LDL was labeled with DiI perchlorate-conjugated LDL (Invitrogen, Waltham, MA, USA), applying a modified procedure of the method described in Teupser et al. [57]. A stock solution of DiI was prepared by dissolving 3 mg DiI in 1 mL DMSO and then was added to the LDL solution to yield a final ratio of 150 μg DiI to 1 mg LDL. Then, it was incubated for 18 h at 37 °C under dark conditions followed by five rounds of centrifugation at 3000–3500× *g* for 45 min, using ultra centrifugal filter units (15 mL; Merck, Darmstadt, Germany) in order to isolate the DiI-labeled LDL. Then, DiI-LDL was dialyzed against saline containing phosphate-buffered saline (PBS) and filter-sterilized (0.25 μm, Water Millex HV units). The ApoB concentrations of LDL and DiI-LDL were determined using commercial kits adapted to a COBAS c501 autoanalyzer (Roche Diagnostics, Minato City, Tokyo). The standard solutions of DiI were prepared in isopropanol with a concentration range of 0–110 ng/mL. The standard curve of DiI-LDL was prepared in saline with a concentration range of 0–1600 ng protein/mL. A spectrofluorometer with excitation and emission wavelengths set at 549 and 564 nm was used to obtain fluorescence measurements. The specific activity of DiI-LDL was finally obtained as the amount of DiI (ng) integrated into 1 μg of LDL.

### 4.4. Analysis of Lipoprotein Uptake by a Fluorometric Assay

LDLR activity and LDL uptake were analyzed measuring the DiI-LDL. Specifically, 3 × 10^4^ cells were seeded in 30 mm dishes with RPMI 1940 (ThermoFisher Scientific, Waltham, MA, USA) and 10% FBS. The day after, the cells were treated with 5% FBS, as a control, and with 200 µg/mL DiI-LDL for 24 h, 48 h and 72 h. For the fluorometric assay, after incubation, the cells were placed on ice and washed three times with cold PBS 1X + 00.4% BSA and twice with cold PBS 1X. Then, 300 μL of lysis reagent (NaOH 0.1 M + 1 g/L SDS) was added and left under consistent shaking at room temperature for 30–60 min. A spectrofluorometer with excitation and emission wavelengths set at 549 and 564 nm was used to measure the fluorescence in 200 μL of the lysate on black microtiter plates. Protein quantification was determined in 10 μL by BCA using BSA dissolved in lysis reagent as a standard (Thermofisher Scientific, Waltham, MA, USA) in order to normalize fluorescence measurements through cellular culture confluence. Additionally, the fluorescence of the DiI-LDL diluted in lysis reagent was measured to determine the specific fluorescence intensity of the DiI-LDL preparation used. 

### 4.5. Analysis of Lipoprotein Uptake by Confocal Microscopy

Specifically, 3 × 10^4^ cells were seeded in 30 mm confocal dishes (VWR, Radnor, PA, USA) with RPMI 1940 and 10% FBS. The day after, the cells were treated with 5% FBS, as a control, and with 200 µg/mL DiI-LDL for 24 h, 48 h and 72 h. Nucleus and lysosomes were labeled with Hoechst and GFP (ThermoFisher Scientific, Waltham, MA, USA) 5 min and 16 h before confocal analysis, respectively. Images of immunostained cells were recorded on a Leica-inverted fluorescence confocal microscope (Leica TCS SP5-AOBS, Wetzlar, Germany). Cells were viewed with HCX PL APO 63X oil/0.6–1.4 objective. Fluorescent images were acquired in a scan format of 1024 × 1024 pixels in a spatial dataset (xyz or xzy) and were processed using the Leica Standard Software TCS-AOBS (V2.7.3). Lisosomal quantification has been carried out using Fiji software (ImageJ V1.53, University of Wisconsin, Madison, WI, USA) [58].

### 4.6. Quantitative Real-Time PCR

Total RNA was isolated using the TRIZOL reagent according to the manufacturer’s instructions (Invitrogen, Waltham, MA, USA). A total of 1 μg of the total RNA was reverse-transcribed using a transcriptor first-strand cDNA synthesis kit (Roche Applied Science, Penzberg, Germany), and the cDNA samples were stored at –20 °C for use as a template in real-time polymerase chain reaction (PCR) analysis. The gene expression profiles were analyzed in an ABI PRISM 7900HF Sequence Detection System, using a predesigned and labeled primer/probe set (Assays-on-Demand™ Gene Expression Assay, Applied Biosystems, Foster City, CA, USA). The Taqman qPCR primers (Applied Biosystems, Foster City, CA, USA) used were as follows: human HMGCR (Hs00168352_m1), human LDL receptor (LDLR; Hs01092524_m1) and human GAPDH (NM_002046.3). All the reactions were performed with 100 ng of cDNA in a total volume of 50 μL of TaqMan^®^ Universal PCR Master Mix (Applied Biosystems, Foster City, CA, USA), and the relative expression levels for each gene were calculated using the 2^−ddCt^ method, with SDS2.3 and Data Assist V2.1 software (Applied Biosystems, Foster City, CA, USA), and GAPDH was used as the normalizing gene.

### 4.7. Protein Extraction and Western Blot 

TPC1 and BCPAP cells were seeded in 60 mm plates and treated with or without LDL (200 µg/mL ApoB) for 24 h, 48 h and 72 h when they reached a 70% of confluence. In terms of inhibitor treatment, the BCPAP cells were treated with 5% FBS and 0.01% DMSO as a control, with LDL (200 μg/mL ApoB) or with vemurafenib (1 μM) (PLX4032, Selleck Chemicals LLC, Houston, TX, USA) for 24 h, 48 h and 72 h. The TPC1 and BCPAP cells were lysed in RIPA buffer (50 mM Tris-HCl, pH 7.5; 150 mM NaCl; 1% NP40; 0.5% sodium deoxycholate; 0.1% SDS; 1 mM EDTA) supplemented with protease inhibitor cocktail (Roche Diagnostics, Minato City, Tokyo), phenylmethylsulfonyl fluoride (PMSF, Sigma, St. Louis, MO, USA) and sodium orthovanadate (Sigma). Lysates were centrifuged at 12,000× *g* for 15 min at 4 °C and a BCA protein assay reagent kit (ThermoFisher Scientific, Waltham, MA, USA) was used to obtain the protein concentrations from the supernatants. Afterwards, the protein extracts were mixed with a 4X Laemmli loading buffer and heated at 94 °C for 4 min. Then, 20 μg of protein was size-separated on a 10% TGX Stain-Free precast gel (Bio-Rad, Hercules, CA, USA), transferred to a 0.2 μm PVDF membrane (Bio- Bio-Rad, Hercules, CA, USA) and the membrane were blocked with 3% dried milk in Tris-buffered saline containing 0.05% of Tween-20 (TBST buffer) for 15 min. Finally, membranes were incubated with optimized dilutions of the primary antibody (Appendix A) overnight at 4 °C. Thereafter, the membranes were washed three times for 10 min with TBST buffer and re-incubated with the IgG HRP-conjugated secondary antibody for 1 h (Appendix A). Finally, the membranes were washed three times for 10 min with TBST buffer and analyzed using an Immun-Star Western Chemiluminescence Kit (Bio-Rad, Hercules, CA, USA). Imaging and data analysis were performed following the protocol described in Taylor et al. and Neris, R.L.S., et al. TGX Stain-free gels were activated for 1 min after SDS-electrophoresis. Images were captured using a ChemiDoc XRS Gel Documentation System (Bio-Rad, Hercules, CA, USA) and Image Lab software (version 6.0.1, Bio-Rad, Hercules, CA, USA). Data normalization analysis for each protein band was performed with the stain-free gel image saved, and the background was adjusted in such a way that the total background was subtracted from the sum of the density of all the bands in each lane [59,60].

### 4.8. MTT Assay

In terms of the MTT assay, 4.000 cells/well were seeded in quintuplicates in 96-well microplates and treated with 5% FBS and 0.01% DMSO as a control, with LDL (200 μg/mL ApoB) or with vemurafenib (PLX4032, Selleck Chemicals LLC, Houston, TX, USA) (1 μM) for 24 h, 48 h and 72 h. The proliferation viability was measured using a 20-µL MTT solution (5 mg/mL) of 3-(4,5-dimethyl-2-thiazolyl)-2,5-diphenyl-2-H-tetrazolium bromide (MTT; Sigma-Aldrich, St. Louis, MO, USA) and incubation at 37 °C for 4 h; then, 50 µL DMSO was added to each well and they were incubated at 37 °C for 10 min. The absorbance at 490 nm was obtained to calculate the cell proliferation rate using a microplate reader (xMark, Bio-Rad, Hercules, CA, USA).

### 4.9. Migration and Adhesion Assay

To analyze the cell migration and adhesion, both cell lines (TPC1 and BCPAP) were seeded at high densities until 70% confluence was reached. In terms of migration capacity, the cells were scratched using a 10 uL pipette tip and washed with PBS 1X. The wounds were photographed at 0 h (t = 0) and after 16 h with or without LDL incubation (200 μg/mL ApoB) and 5% FBS at 37 °C using an inverted microscope and analyzed with Image J software V1.53 (University of Wisconsin, Madison, WI, USA) [58]. The percentage of wound-healing was obtained from a minimum of three measurements of the wound area, and each result was the mean of three independent experiments. In terms of the adhesion assay, the cells were treated with or without LDL (200 μg/mL ApoB) and 5% FBS and then incubated for 24 h, 48 h and 72 h. Afterwards, the percentages of viable suspension cells in the supernatant were counted using the automated Cell Counter (Bio-Rad, Hercules, CA, USA) together with trypan blue exclusion assay for a direct identification and enumeration of live (unstained) and dead (blue) cells in a given population.

### 4.10. Transient Transfection Assay

To knockdown LDLR expression, a siRNA transfection in BCPAP cell line was performed via JetPrimeTM (Polyplus transfection, VWR, Radnor, PA, USA), according to the manufacturer’s instructions. Approximately 1.5 × 10^5^ cells were seeded in 30 mm plates for protein and RNA extraction one day prior to transfection. Cells were transfected with either 80 nM pre-designed short, interfering RNA for human LDLR (siLDLR) (IDs4) (Ambion, Life Technologies, Carlsbad, CA, USA) or control siRNA (Mock-siRNA) purchased from Santa Cruz Biotechnology, Inc. (sc-37007) (Dallas, TX, USA) in RPMI 1940 medium (ThermoFisher Scientific, Waltham, MA, USA) supplemented with 10% FBS, 100 U/mL penicillin and 1 μg/mL streptomycin. After 48 h, the transfection media was replaced with RPMI 1940 medium (ThermoFisher Scientific, Waltham, MA, USA) supplemented with 5% FBS, 100 U/mL penicillin and 1 μg/mL streptomycin, the cells were transfected again with the corresponding siRNA (siLDLR or Mock-siRNA) and treated for 48 h with 5% FBS and 0.1% DMSO, as a control; LDL (200 µg/mL ApoB) and 1μM of vemurafenib (PLX4032, Selleck Chemicals LLC, Houston, TX, USA), when corresponds. The 1 μM dose of vemurafenib (PLX4032, Selleck Chemicals LLC, Houston, TX, USA) was chosen in accordance with previous studies in BCPAP cell lines in order to just modulate the RAS/RAF/MAPK (MEK)/ERK signaling pathway without promoting cellular toxic effects [61,62].

The following steps after siRNA transfection for lipoprotein uptake analysis by fluorometric assay and confocal analysis in the BCPAP cell line are described in Section 4.4 and Section 4.5, respectively. 

For MTT assay (MTT; Sigma-Aldrich, St. Louis, MO, USA), 3 × 10^3^ cells were plated in 96-well plate and transiently transfected with 5nM siRNA LDLR or Mock-siRNA, following the same steps described above.

### 4.11. Statistical Analysis

GraphPad Prism version 9.0 (GraphPad Inc., San Diego, CA, USA), with a *p*-value < 0.05 denoting statistical significance, was used for the statistical analysis. The two-way ANOVA test as well as Sidak’s multiple comparisons test were used to evaluate the effects of time and cell type on each dependent variable. The analysis of more than two groups was performed via one-way ANOVA with Tukey’s post hoc test where applicable. Differences between the two groups were analyzed via an unpaired two-tailed Student’s *t*-test.

## Figures and Tables

**Figure 1 ijms-24-11153-f001:**
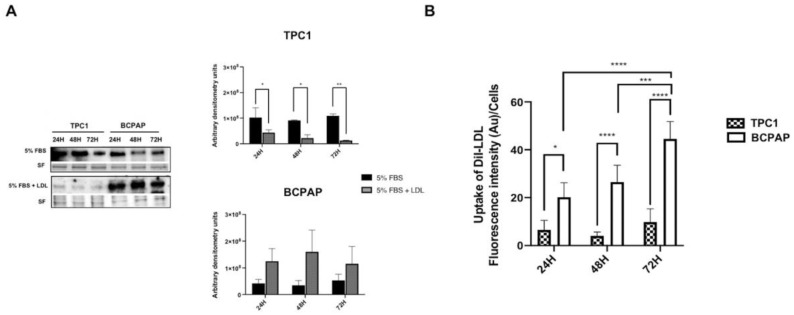
Low-density lipoprotein (LDLR) protein expression and 19-dioctadecyl-3,3,39,39-tetramethyl indocarbocyanine (DiI)-low-density lipoprotein (LDL) uptake in TPC1 and BCPAP cell lines. (**A**) Cells were treated with basal conditions (5% FBS) or LDL (200 μg/mL ApoB) and harvested at 24 h, 48 h and 72 h before analysis. Left panel: one representative blot is shown, and stain-free gel (SF) was used as the loading control. Graphs show densitometry of the Western blots relative to basal condition-treated cells (5% FBS) in comparison to LDL-treated cells. (**B**) Cells were exposed to DiI-LDL (200 μg/mL ApoB) for 24 h, 48 h and 72 h before analysis for mean fluorescence intensity by fluorescence spectrometer to compare LDL uptake between both cell lines. Statistical analysis: a two-way ANOVA test plus Sidak’s multiple comparisons test were performed to compare the LDLR protein expression and DiI-LDL uptake between the two groups at each time point (* *p* < 0.05, ** *p* < 0.002, *** *p* = 0.0003, **** *p* < 0.001). Data are expressed as mean ± SEM of a minimum of three independent experiments (N = 3).

**Figure 2 ijms-24-11153-f002:**
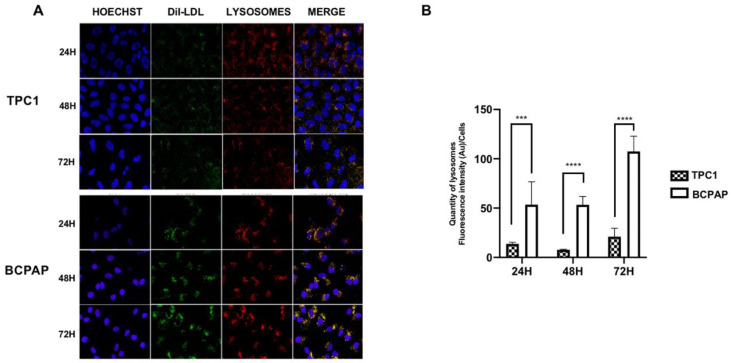
Confocal images of DiI-LDL uptake and lysosome quantification. (**A**) Confocal microscopy of Hoechst-stained (blue), DiI-LDL-stained (green) and lysosome-GFP (red) in TPC1 and BCPAP cell lines. Cells were treated for 24 h, 48 h and 72 h with DiI-LDL (200 μg/mL ApoB). The presence of colocalization of the red and green signals in the merged images is highlighted in yellow. The scale bar size was set to 50 µm. (**B**) Quantity of lysosomes in TPC1 and BCPAP cell lines after DiI-LDL (200 μg/mL ApoB) incubation for 24 h, 48 h and 72 h. Statistical analysis: a two-way ANOVA test plus Sidak’s multiple comparisons test were performed to compare the lysosome quantification between the cell lines (*** *p* < 0.0009 and **** *p* < 0.0001). Data are expressed as mean ± SEM of three independent experiments (N = 3).

**Figure 3 ijms-24-11153-f003:**
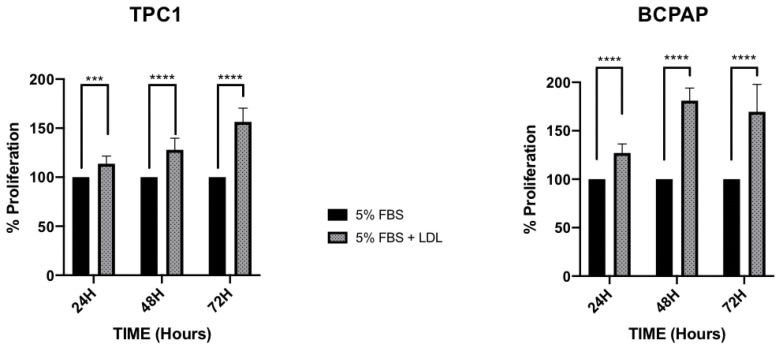
Percentage of cellular proliferation determined by MTT assay of the TPC1 and BCPAP cell lines. Cells were treated with basal conditions (5% FBS), as a control, or with LDL (200 μg/mL ApoB) for 24 h, 48 h and 72 h. Statistical analysis: a two-way ANOVA test plus Sidak’s multiple comparisons test were performed to compare both groups. (*** *p* < 0.0009, **** *p* < 0.001). Data are expressed as mean ± SEM of three independent experiments (N = 3) carried out in quintuplicate.

**Figure 4 ijms-24-11153-f004:**
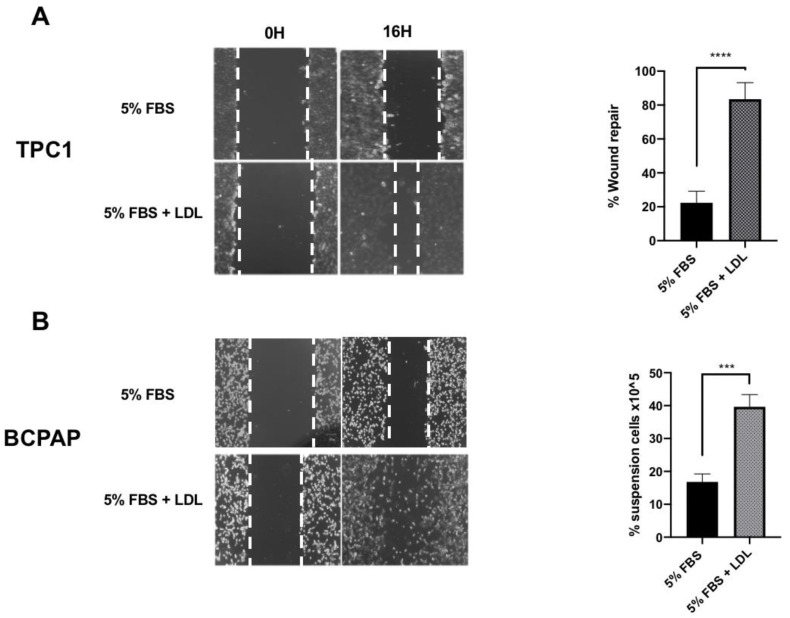
Effects of LDL on cellular migration. A line was scratched in both the TPC1 and BCPAP cell lines, and cultures were treated with 5% FBS, as a control, or with LDL (200 µg/mL ApoB) for 16 h. (**A**) The graph represents the percentage of wound-healing repair (**** *p* < 0.0001) in the TPC1 cell line treated with LDL (200 µg/mL ApoB) compared to the control at basal conditions (5% FBS). (**B**) The graph represents the percentage of suspension BCPAP cells treated with LDL (200 µg/mL ApoB) compared to the control at basal conditions (5% FBS). Images are at 10× resolution. Statistical analysis: an unpaired *t*-test was performed to compare the LDL-treated cells with the control condition (*** *p* = 0.0009, **** *p* < 0.0001). The results are presented as the mean ± SEM of three independent experiments.

**Figure 5 ijms-24-11153-f005:**
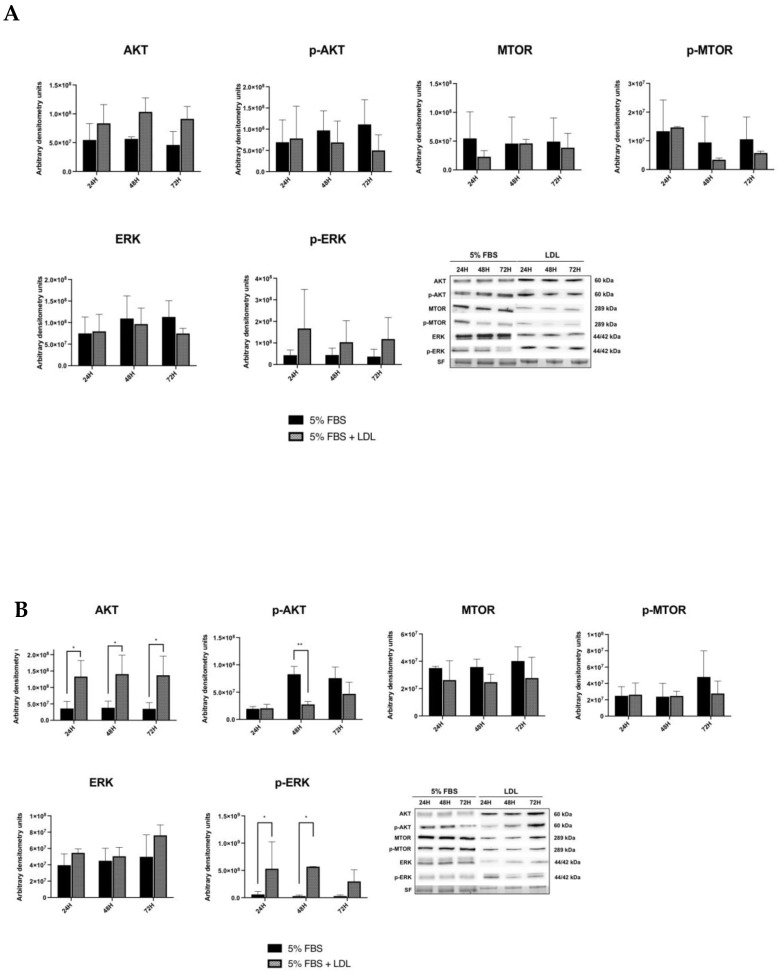
Western blot analysis for PI3K–AKT–MTOR and RAS/RAF/MAPK (MEK)/ERK pathways. (**A**) TPC1 cells were treated with basal conditions (5% FBS) or LDL (200 μg/mL ApoB) and harvested at 24 h, 48 h and 72 h. Last panel: representatives blots are shown. Stain-free (SF) gel was used as the loading control. Graphs show the densitometry of the Western blots relative to basal condition-treated cells in comparison to LDL-treated cells. (**B**) BCPAP cells were treated with basal conditions (5% FBS) or LDL (200 μg/mL ApoB) and harvested at 24 h, 48 h and 72 h. Last panel: representative blots are shown. Stain-free (SF) gel was used as the loading control. Graphs show densitometry of the Western blots relative to basal condition-treated cells in comparison to LDL-treated cells. Statistical analysis: two-way ANOVA test plus Sidak’s multiple comparisons test (* *p* < 0.01, ** *p* < 0.001). Data are expressed as mean ± SEM of a minimum of three independent experiments (N = 3).

**Figure 6 ijms-24-11153-f006:**
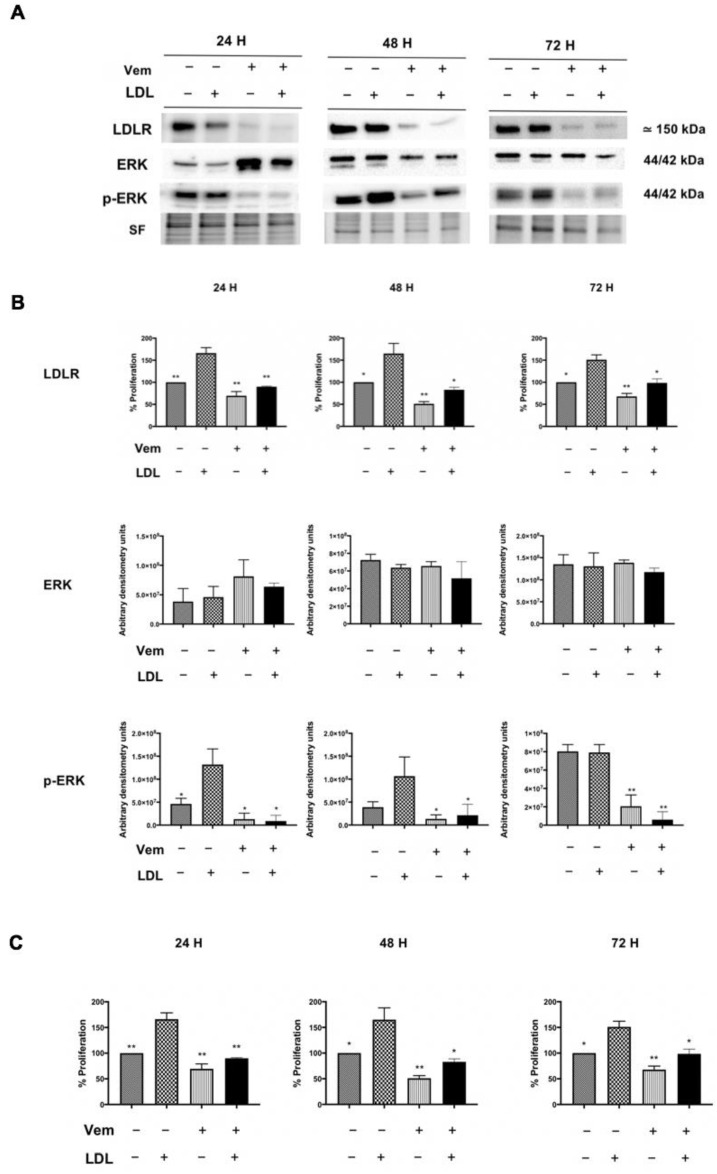
Western blot analysis of Ras/Raf/MAPK (MEK)/ERK pathway and cellular proliferation determined by MTT assay of BCPAP cell line after LDL (200 µg/mL ApoB) incubation and/or vemurafenib treatment (1 µM) compared to the LDL-only treatment condition (200 µg/mL ApoB + DMSO 0.1%) for 24 h, 48 h and 72 h. (**A**) Representative blots of LDLR, ERK and p-ERK are shown. Stain-free (SF) gel was used as the loading control. (**B**) Graphs show the densitometry analysis of the Western blots of LDLR, ERK and p-ERK after LDL (200 µg/mL ApoB) incubation and/or vemurafenib treatment (1 μM) for 24 h, 48 h and 72 h in comparison to the LDL-only treatment condition (200 µg/mL ApoB + DMSO 0.1%). (**C**) Proliferation percentage determined by MTT assay in the BCPAP cell line after incubation with LDL (200 µg/mL ApoB) and/or vemurafenib treatment (1 µM) compared to the LDL-only treatment condition (200 µg/mL ApoB + DMSO 0.1%) at 24 h, 48 h and 72 h. Statistical analysis: one-way ANOVA test plus Tukey’s multiple comparisons test (* *p* < 0.01, ** *p* < 0.001 vs. the LDL-alone condition). All data are expressed as mean ± SEM of three independent experiments (N = 3). Each MTT assay experiment was carried out in quintuplicate.

**Figure 7 ijms-24-11153-f007:**
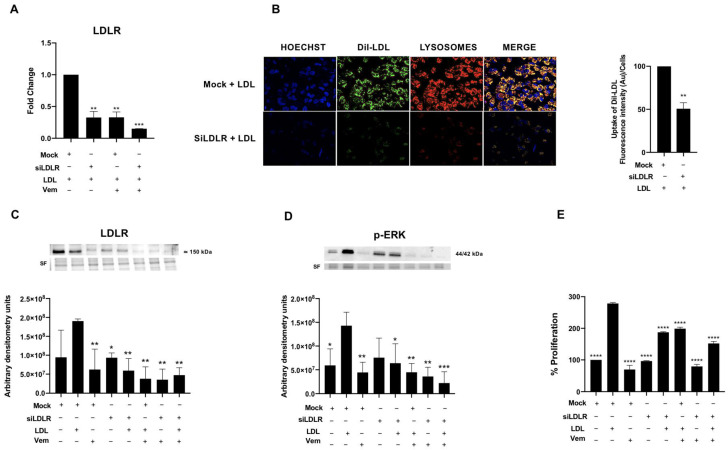
*LDLR* mRNA levels, confocal images and analysis of DiI-LDL uptake; LDLR and p-ERK expression; and cellular proliferation after siLDLR, vemurafenib (1 µM) and/or LDL (200 µg/mL ApoB) treatment compared to basal conditions for 24 h. (**A**) mRNA expression of *LDLR* after siLDLR for 48 h, LDL (200 µg/mL ApoB) and/or vemurafenib (1 µM) treatment in comparison to the LDL-only condition (LDL + DMSO 0.1% + Mock). (**B**) Confocal microscopy of Hoechst-stained (blue), DiI-LDL-stained (green) and lysosome-GFP (red) in BCPAP cell line after 24 h of DiI-LDL (200 µg/mL ApoB) treatment and with and without siLDLR for 48 h. The presence of colocalization of the red and green signals in the merged images is highlighted in yellow. The scale bar size was set to 50 µm. Right graph: cells were exposed to DiI-LDL (200 μg/mL ApoB) for 24 h after siLDLR for 48 h in comparison to the LDL-only condition (Mock + LDL) and analyzed for mean fluorescence intensity by fluorescence spectrometer. Statistical analysis: an unpaired *t*-test plus Welch’s correction were performed to compare both conditions (** *p* = 0.066). Data are expressed as mean ± SEM of a minimum of three independent experiments (n = 3). (**C**) Representative blot of LDLR is shown. Stain-free (SF) gel was used as the loading control. Graph shows the densitometry analysis of the Western blots of LDLR after siLDLR, LDL (200 µg/mL ApoB) incubation and/or vemurafenib treatment (1 μM) for 48 h, in comparison to the LDL-alone condition (LDL + DMSO 0.1% + Mock). (**D**) Representative blot of p-ERK is shown. Stain-free (SF) gel was used as the loading control. Graph shows the densitometry analysis of the Western blots of p-ERK after siLDLR, LDL (200 µg/mL ApoB) incubation and/or vemurafenib treatment (1 μM) for 48 h, in comparison to the LDL-only condition (LDL + DMSO 0.1% + Mock). (**E**) Proliferation percentage determined via MTT assay after incubation with LDL (200 µg/mL ApoB), vemurafenib treatment (1 µM) and/or siLDLR, compared to the LDL-only condition (LDL + DMSO 0.1% + Mock) at 48 h. Statistical analysis: one-way ANOVA test plus Tukey’s multiple comparisons test (* *p* < 0.01, ** *p* < 0.001, *** *p* = 0.0009, **** *p* < 0.0001 vs. the LDL-only condition). All data are expressed as mean ± SEM of three independent experiments (N = 3). Each MTT assay experiment was carried out in quintuplicate.

**Figure 8 ijms-24-11153-f008:**
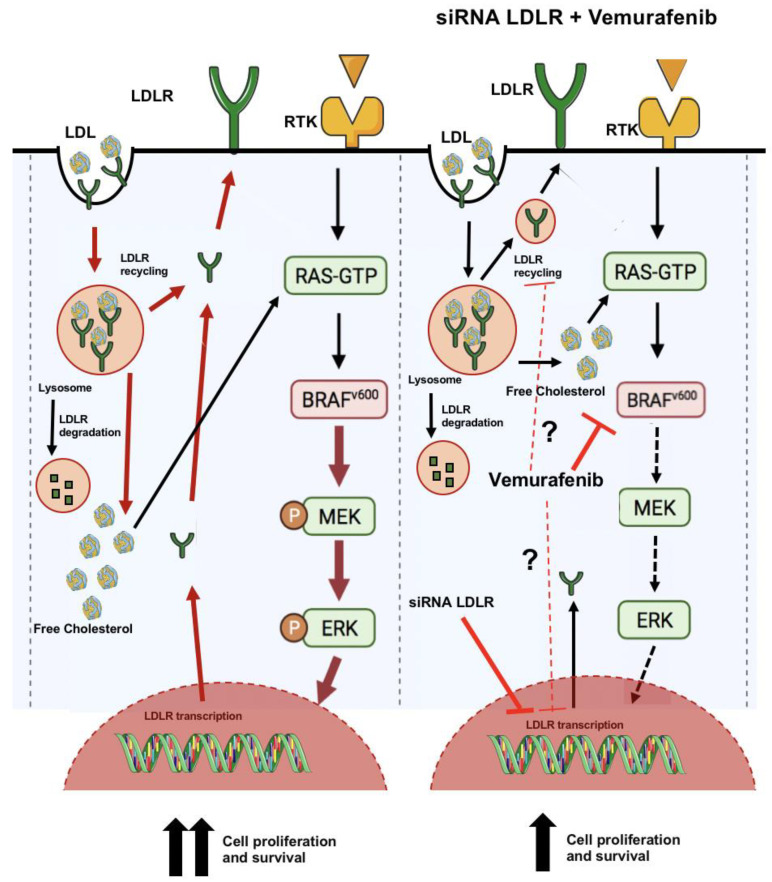
Schematic illustration of the hypothesis that proposes a crosstalk between RAS/RAF/MAPK (MEK)/ERK pathway and LDL-mediated receptor uptake in BCPAP cell line. In the presence of LDL, a cell line harboring a BRAF mutation (BCPAP) can increase proliferation and worsens its behavior by increasing the activation of the RAS/RAF/MAPK (MEK)/ERK pathway. Vemurafenib treatment at 1uM, which enters to the cell by passive diffusion, interrupts the B-Raf/MEK step, as well as downregulating LDLR expression. Additionally, siRNA LDLR downregulates LDLR expression, decreasing LDL uptake and modulating RAS/RAF/MAPK (MEK)/ERK pathway overactivation. Black arrows mean the normal activation of the pathway, red arrows are used when the pathway is overactivated and dashed arrows symbolize when the pathway is inhibited. Part of the Servier Medical Art by Servier is licensed under a Creative Commons Attribution 3.0 Unported License (https://smart.servier.com/image-set-download/ (accessed on 9 November 2022)).

## Data Availability

Not applicable.

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
