# Peer review of "Low-Density Lipoprotein Receptor Is a Key Driver of Aggressiveness in Thyroid Tumor Cells"

_ijms, 2023, doi:10.3390/ijms241311153_

Round 1
Reviewer 1 Report (New Reviewer)
Overall, this study provides valuable insights into the interplay between LDL, LDLR, the MAPK pathway, and the aggressiveness of PTC. The findings suggest potential avenues for targeted therapy in PTC, emphasizing the importance of considering individual cell line characteristics and the presence of BRAF V600E mutation in treatment strategies. However, there are several important issues that should be discussed regarding this study
1. One limitation of this study is the use of only two thyroid cell lines, TPC1 and BCPAP, to investigate the effects of LDL and its receptor on the aggressiveness of PTC. A crucial aspect that should be addressed is the lack of a control group for comparison in terms of LDLR expression. Without a control group, it becomes challenging to assess the significance of LDLR deregulation observed in the experimental groups.
Author Response
Please see the attachment

Reviewer 2 Report (New Reviewer)
This paper entitled “Low-density lipoprotein receptor is a key driver of affressiveness in thyroid tumor cells” by G. Revilla describes how LDL or LDL receptor (LDLR) expression is associated with activation of the MAP kinase pathway, an intracellular growth pathway, in two cell lines with and without BRAF mutations. This is a solid paper and is analyzed using legitimate methods. Depending on the specificity of the cell line, with or without BRAF mutation, the results of this paper reveal a relationship between the LDL-signaling cascade and the regulation of the RAS/RAF/MAPK/ERk pathway. There are no issues to be noted regarding the research design, methodology, results obtained, and interpretation of the results in this paper. Further studies are awaited to see if similar results can be obtained not only in cell lines but also in clinical samples.
Round 2
Reviewer 1 Report (New Reviewer)
I appreciate your response.
This manuscript is a resubmission of an earlier submission. The following is a list of the peer review reports and author responses from that submission.
Round 1
Reviewer 1 Report
2023-0331-IJMS: ‘Role of Braf v600e Mutation in LDL-Mediated Signaling Pathways Increases Aggressiveness in Thyroid Tumor Cells’ by Revilla et al.
This manuscript aims to unravel a mechanistic link between cellular dynamics of the LDL/LDLR pathway to the cell biology of thyroid tumor cells. Specifically, the authors aim to investigate whether the BRAF V600E mutation modifies signaling pathways modulated by the LDL uptake in TPC1 and BCPAP papillary cell lines (WT and BRAF V600E, respectively) co-cultured with LDL and the concomitant effects on RAS-RAF-MAPK signaling.
While the overall hypothesis is clearly stated, the manuscript falls short of the stated goals. The manuscript is difficult to read, lacks rigor in experimental design, data analyses, and data presentation, which together diminish the significance of the findings. The methods used lack sufficient details to ascertain the robustness of the studies. The figures lack clarity in labeling and many of the histograms incorrectly label the axes as %-Proliferation, etc., when in fact they are described in the text as ratios (which are unit-less numbers).
The authors do not elaborate on the inconsistency in the results of Fig. 1B (showing LDL uptake) despite a dramatic loss of LDLR expression (when both cell lines TPC1 & BCPAP are compared). Why are uptake numbers not a direct reflection of LDLR-#? In Fig. 1A, no LDLR expression profiles are shown for untreated cells (only 5% FBS) over the 72h time-period to ascertain ‘intrinsic noise’ in this system. The authors state (line 77 & 78) that “Global analysis of the LDLR expression between the two cell lines detected a significant variation factor of 55.37% (Figure 1A) and a significant variation factor of 16.38% for their uptake obtained through fluorescence analysis (Figure 1B)”. It is unclear what the authors mean by “Global analysis” or by “significant variation factor”, and the latter are reported as % values with ‘4 Significant Figures’, a level of precision that is not generally applicable to biological analysis (of this type). In Fig. 2, the authors note (line 120) “data are expressed as mean ± SEM (n=3)”, which appears to be a sample replicate & is missing an ‘experimental replicate’. In Fig. 3, it is unclear what ‘% proliferation’ means, and a close reading of the figure legend seems to imply that the values reported are a ratio of ‘with/without LDL compared to control cells’, and if so, are ‘unit-less numbers’ not % proliferation. Similar problems affect the remainder of the studies (Figs. 4, 5 & 6). Finally, the studies provide no evidence for a direct connection of LDLR to the RAS-MAPK as pathway, as modeled in Fig. 7.
These (and other) shortcomings indicate that the authors’ conclusions are not supported by the available data. Thus, the manuscript does not provide clear and substantial evidence to support the underlying hypothesis that LDL-LDLR dynamics directly impact proliferative potential via the RAS-RAF-MAPK pathway. For these reasons, the manuscript is not considered acceptable for publication in IJMS.
Reviewer 2 Report
In this research article, the authors demonstrated the role of BRAF V600E mutation in modulating the LDL-LDLR axis using BCPAP cells harboring the BRAF V600E mutation and a thyroid tumor cell line TPC1. The authors demonstrated the different LDLR expression, LDL-induced proliferation, and changes in the PI3K/AKT and MEK signaling pathways, and further demonstrated that the change can be countered by BRAF inhibition by vemurafenib.
Overall, the study is well-designed, the manuscript is written clearly, and the results are timely and relevant for thyroid cancer therapeutics. Please see below my comments.
Specific Comments:
1. A description of the two cell lines BCPAP and TPC1 used for the analysis should precede the discussions about the results obtained using these cell lines. It seems that the authors mentioned this later on in the results section (“BCPAP (harboring the BRAF v600e mutation” …).
2. Are the “non-phospho ERK” measured by antibody “Cell signaling #9102” indicated in the supplement? If so, doesn’t it measure total ERK and not “non-p-”? Please verify and elaborate how this is done.
3. Related to the previous point, in Figure 6A, is the ERK band reporting non-phospho or total ERK? Please elaborate. Does the expression of total ERK change with LDL simulation and vemu?
4. It would be helpful to briefly explain how the dose of vemurafenib is chose. What is the EC50? Do the authors expect a dose-response?
5. Fig. 7 seems to have some text on the left cropped out.
6. In the supplement, the antibody for ERK1/2 is labeled “mouse” – I assume that’s a misprint and should be rabbit?
7. Please spell out all the abbreviations upon their first appearances in the article.
Reviewer 3 Report
The manuscript present data comparing two different cell lines, WT and V60E mutated, claiming differences due to the presence of the mutation. The authors should manipulate BRAF expression using crisp and/or overexpression strategies to compare the same cell line in different conditions, otherwise the results don't have scientific impact and the conclusions are not fully supported by the data